## RESEARCH ARTICLE

# Human adipose stromal cells differentiate towards a tendon phenotype with adapted visco-elastic properties in a 3D-culture system

Maxime Hordé[1], Jonathan Fouchard[1], Luna Gomez Palacios[1], Xavier Laffray[2], Cédrine Blavet[1], Véronique Béréziat[3], Claire Lagathu[3], Ludovic Gaut[1], Delphine Duprez[1,*,‡] and Emmanuelle Havis[1,*,‡]

## ABSTRACT

Tendon cell differentiation relies on molecular and mechanical parameters that control the expression of tendon-associated transcription factors and extracellular matrix proteins. However, the minimal cues able to initiate tendon differentiation from progenitor cells remains unknown. We analysed the tendon differentiation program in human adipose stromal cells (hASCs) cultured in a minimal 3D system. We generated 3D-hASC constructs by embedding hASCs in a type-I collagen gel under a static uniaxial geometrical constraint with no additional molecular and mechanical cues, and assessed tendon-associated gene expression and mechanical properties for up to 3 weeks in culture. Analysis of tendon-associated genes revealed a molecular progression consistent with the acquisition of a tendon phenotype. The analysis of viscoelastic properties of 3D-hASC constructs by nano-indentation indicated a progressive increase in tissue stiffness up to 10 kPa, concomitant with a reduced stress relaxation indicative of solid-like mechanical properties. These changes in mechanical properties parallel the molecular change of matrix genes during the time of cultures. In summary, we have established that hASCs cultured in a minimal 3D-system progress into the tendon differentiation program associated with variations of mechanical properties.

KEY WORDS: Tendon differentiation program, Tendon-associated matrix, Collagen hydrogel, Viscoelastic properties, Human adipose stromal cells, 3D-culture system

## INTRODUCTION

The tendon is a key component of the musculoskeletal system that transmits forces from muscle to bone to allow locomotion. To accomplish this function, tendons exhibit specific viscoelastic properties, with a Young's modulus much higher than other soft tissues (Peltonen et al., 2013; Connizzo and Grodzinsky, 2017). Tendon is a connective tissue composed of tendon fibroblasts that produce a dense extracellular matrix (ECM) consisting mainly of type I collagen fibrils that display a specific spatial organisation to resist the tensile forces generated by muscle contraction. Compared to other components of the musculoskeletal system, little is known about the mechanisms involved in tendon cell differentiation from undifferentiated cells. There is no identified master gene for the tendon lineage, i.e. there is no identified gene whose expression alone can trigger the tendon differentiation program; this stands in contrast to the controlling role of MYOD in the muscle program (Tapscott, 2005). In addition, the tendon program is also difficult to follow because of the lack of recognised markers for each step of the tendon differentiation program (Huang and Galloway, 2023). However, tendon fibroblast populations have been recently characterised with single-cell RNA-sequencing technology in developing chicken and mouse limbs (Hirsinger et al., 2024; Arostegui et al., 2022; Coren et al., 2024).

In vertebrates, type I collagen, encoded by *COL1A1* and *COL1A2* genes, is the main structural and functional component of tendons. The regulation of *COL1A1* transcription is under the control of several transcription factors at different steps of tendon formation. The main developmental tendon marker, the bHLH transcription factor Scleraxis (SCX) (Schweitzer et al., 2001) regulates *COL1A1* transcription during development (Murchison et al., 2007). Two other transcription factors have been shown to regulate *COL1A1* transcription in adult mouse tendons, EGR1 (Guerquin et al., 2013) and MKX (Mohawk) (Ito et al., 2010; Liu et al., 2010). In addition to *COL1A1* regulation, SCX is required and sufficient for the expression of the transmembrane glycoprotein *TNMD* (Tenomodulin) in developing limb tendons and tendon cell cultures (Shukunami et al., 2006, 2018; Murchison et al., 2007); TNMD being involved in tenocyte proliferation and tendon collagen fibril maturation in newborn mice (Docheva et al., 2005; Dex et al., 2017). This indicates that SCX acts as a starter of the tendon program, although not being a master gene for the tendon lineage. Tendons are still observed in *Scx* mutant mice; however, force-transmitting tendons display severe alterations with disorganised matrix (Murchison et al., 2007; Huang et al., 2019). The transcriptomic analysis of *SCX*-positive cells during mouse limb tendon development provided us with a list of genes enriched in foetal tendon cells (Havis et al., 2014). Although the function of all the 100 top genes have not been studied, when studied, gene function analyses show an involvement in tendon collagen fibrillogenesis; exampled with *THBS2*, coding for Thrombospondin 2 a secreted ECM glycoprotein (Kyriakides et al., 1998), *COL6A3* encoding the chainα3 of the non-fibrillar type VI collagen (Pan et al., 2013) and *COL14A1* encoding the chainα1 of the FACIT (fibril-associated

[1]Sorbonne Université, Institut Biologie Paris Seine, CNRS UMR7622, Developmental Biology Laboratory, Inserm U1156, F-75005 Paris, France. [2]Université Paris Est Creteil, Glycobiology, Cell Growth and Tissue Repair Research Unit (Gly-CRRET), EA 4397, F-94010 Creteil, France. [3]Sorbonne Université, Inserm UMRS938, Centre de Recherche Saint-Antoine (CRSA), Institut Hospitalo-Universitaire de Cardio-Métabolisme et Nutrition (ICAN), F-75012 Paris, France. *Co-senior authors

‡Authors for correspondence (delphine.duprez@sorbonne-universite.fr; emmanuelle.havis@sorbonne-universite.fr)

J.F., 0000-0002-9976-462X; L.G.P., 0009-0000-0692-8828; X.L., 0000-0001-5917-9154; C.B., 0000-0001-8584-973X; V.B., 0000-0002-9795-549X; C.L., 0000-0003-0700-8286; L.G., 0000-0003-3987-3389; D.D., 0000-0003-0248-7417; E.H., 0000-0001-5634-3174

collagen with interrupted triple helices) (Ansorge et al., 2009). The top 100 genes also include various ECM proteins contributing to tendon biomechanical properties, such as the small leucine-rich proteoglycan, DCN (Decorin) (Zhang et al., 2006), the secreted matrix protein POSTN (Periostin) (Li et al., 2023; Ackerman et al., 2024), the collagen-bonding protein, DPT (Dermatopontin) (Okamoto and Fujiwara, 2006; de Micheli et al., 2020; Omoto et al., 2022) and the lysyl oxidase enzyme LOX, which catalyses collagens cross-linking of through oxidative deamination of lysine residues (Nguyen et al., 2023). All these tendon-associated molecules provide us with molecular markers to assess the developmental tendon phenotype. In addition to molecular cues, mechanical parameters are key for tendon development. Axial, head and limb tendon development is stopped in the absence of muscle contraction in chicken and mouse models (Gaut and Duprez, 2016; Havis et al., 2016).

Tendon cell differentiation is difficult to follow *in vitro*, since the tendon phenotype is not maintained in tendon 2D-cell cultures. Stem cells require molecular stimuli to induce a tendon phenotype when cultured in 2D (Guerquin et al., 2013; Otabe et al., 2015; Gaut et al., 2020). Compared to 2D-culture systems, 3D-culture systems are more favourable to trigger tendon differentiation (Yang et al., 2013; Gaut et al., 2020; Jaiswal et al., 2020). Human, rat, mouse and avian tendon fibroblasts have been extensively used in 3D-culture systems to study collagen fibrillogenesis (Kapacee et al., 2008; Bayer et al., 2010; Kalson et al., 2010; Herchenhan et al., 2013, 2015; Avey et al., 2024). In a fibrin 3D-culture system, chicken embryonic tendon cells were able to produce matrix genes resembling to native tendons (Yeung et al., 2015).

Mesenchymal stromal cells (MSCs) display progenitor-like properties and are capable of differentiating into different cell lineages, including osteoblasts, adipocytes or chondrocytes. The use of MSC allows the study of the mechanisms by which the tendon lineage is initially selected (Jaiswal et al., 2020). The tendon differentiation potential has been analysed in numerous MSC types, such as bone marrow-derived mesenchymal stromal cells (bmMSCs), human adipose-derived stromal cells (hASCs), mouse MSCs (with the C3H10T1/2 cell line). However, these 3D-culture systems made of MSCs require molecular cues (Yang et al., 2013; Hsieh et al., 2018; Kapacee et al., 2010; Guerquin et al., 2013; Gaut et al., 2016) or mechanical loading (Kuo and Tuan, 2008; Scott et al., 2011; Guerquin et al., 2013; Park et al., 2022; Pentzold and Wildemann, 2022) to reach and maintain a tendon phenotype.

Here, we used a 3D-culture system with hASCs in a type I collagen hydrogel maintained between two anchor points to assess whether hASCs were able of undergoing a tendon differentiation program with no external molecular cues or mechanical stimulation for 3 weeks. The activation of tendon-associated gene expression (transcription factors, matrix and other tendon markers) combined with increased stiffness and reduced relaxation of 3D-hASC constructs show that hASCs are directed towards a tendon differentiation program. This shows that hASCs undergo towards a tendon phenotype in a minimal 3D-culture system with no external molecular or mechanical cues.

## RESULTS
### Development of a minimal 3D-culture system with hASCs
Since adipose tissue is an attractive alternative source of MSCs (Zuk, 2013; Kokai et al., 2014), we chose hASCs isolated from the stromal vascular fraction of human adipose tissue collected after liposuction surgery (Béréziat et al., 2019; Gorwood et al., 2020) to analyse how MSCs self-organise and differentiate when cultured in 3D-conditions. During embryonic development, tendon primordia are high-cell-density linear structures anchored between skeletal and muscle tissues.

To mimic the tendon biophysical microenvironment, hASCs were homogenously embedded in a collagen hydrogel, which reproduces the native microenvironment of tendon tissues (Garvin et al., 2003; Mousavizadeh et al., 2023), deposited between two anchor points using the Flexcell technology (Fig. S1). This technology consists of a holder with 24 cylindrical moulds conferring a homogeneous and regular shape to the constructs. Each mould is perforated with vertical holes through which the vacuum is applied, allowing the silicon elastic surface of the well to acquire the shape of the mould (Fig. 1A). At the extremities of each cylinder, two nylon anchor stems allowed the maintenance of static tension in each construct (Fig. 1A). After isolation and amplification, hASCs were embedded in a 3.5% bovine type I collagen solution and seeded between the two anchor stems. Vacuum was maintained for 2 h to generate a construct of a cylindrical shape. 3D-hASC constructs (which we define as Day −2) were then left for 48 h to allow cell development under static tension by condensing the surrounding matrix. 3D-constructs were analysed starting from this time point, defined as Day 0 (Fig. 1B). Once generated, the 3D-constructs were maintained under static conditions for 21 days (Fig. 1B). We did not apply external forces and the 3D-constructs were only subjected to the tension forces exerted between the two anchoring points. At Day 0, 3D-hASC constructs exhibited diameter values of 2.08±0.08 mm (Fig. 1D). The diameters of 3D-hASC constructs decreased drastically from Day 0 to Day 7 (0.67 ±0.02 mm at Day 7) and then displayed a slow decrease from Day 7 to Day 21 (0.59±0.02 mm at Day 14, 0.55±0.04 mm at Day 21) (Fig. 1D). Notably, because the length of the construct is fixed, we could estimate a volume reduction of 3D-constructs of about 15-fold in the course of the 21 days of culture. 3D-no cell constructs not seeded with hASCs exhibit constant diameter values during the 3 weeks of culture, 4.46±0.48 mm at Day 0, 4.41±0.45 mm at Day 7, 4.5±0.45 mm at Day 14 and 4.46±0.48 mm at Day 21 (Fig. S1D-E). Similarly, the cross-sectional areas of 3D-no cell constructs remain constant and significantly higher than those of 3D-hASC constructs (Fig. S1F). The total number of cells in the constructs remained relatively stable between Day 0 and Day 21, with a slight increase at Day 7 and a slight decrease between Day 7 and Day 21 (Fig. 1D). Consistently, we observed very few apoptotic and proliferative cells visualised by cleaved-Caspase 3 and Phospho-Histone H3 immunostaining, respectively, in 3D-hASCs at different time points between Day 0 and Day 21 (Figs S2 and S3).

At a cellular level, hASCs in this 3D-culture system displayed a fusiform shape and spindle-shaped nuclei, aligned in the direction defined by the two anchor points (Fig. 1E), reminiscent of cell organisation in native tendons, while hASCs exhibited a star shape in 2D cultures (Fig. S1B). In addition, cells progressively aligned along the tension axis as the angle between the cell axis and the axis of the 3D-hASC constructs decreased throughout culture (Fig. 1F). RT-qPCR analysis of the expression of the main tendon collagen *COL1A1* show a progressive increase from Day 0 to Day 21 (Fig. 1G). *In situ* hybridisation revealed that *COL1A1* was highly expressed in tendon constructs until Day 21 (Fig. 1H-I). However, at Day 21, *COL1A1* appeared more expressed in central than peripheral regions (Fig. 1I). Immunofluorescence staining confirmed the presence of collagen I fibres oriented along the construct axis in 3D hASCs (Fig. S4). We used this 3D-culture system to analyse the tendon differentiation potential of hASCs.

### hASCs are prone to tendon differentiation in a minimal 3D-culture system
In order to assess the tendon differentiation potential of hASCs cultured in the 3D-conditions described above, we analysed tendon

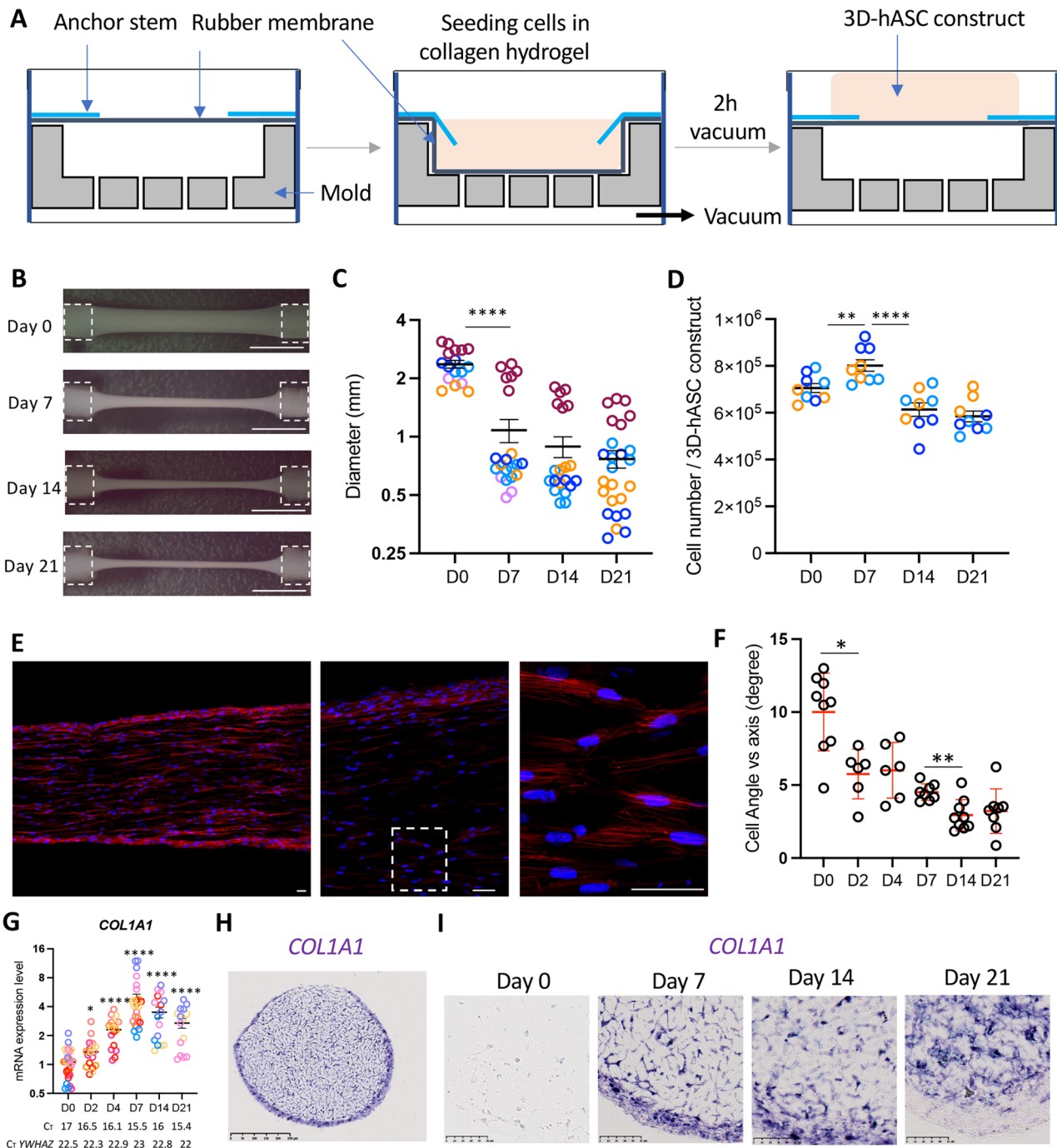

**Fig. 1. Generation of 3D-hASC constructs using a Flexcell bioreactor.** (A) Human adipose stromal cells are isolated from lipoaspirates and amplified before seeding. The cells are embedded in a 3.5% collagen hydrogel and deposited between the two anchor points. The polymerisation is performed by maintaining the vacuum and at 37°C for 2 h. The vacuum is then broken and the 3D-hASC constructs are kept in the incubator for 48 h before starting the experiments. Scale bar: 100 µm. (B) 3D-hASC constructs were cultured for 21 days and (C) diameters were measured from constructs at Day 0 (*n*=11), Day 7 (*n*=15), Day 14 (*n*=14) and Day 21 (*n*=19). Each colour represents a set of experiments. Four independent experiments were performed with 3<*n*<8 biological replicates for each experiment. The *P*-values were obtained using the Mann–Whitney test. Asterisks * indicate the *P*-values of 3D-hASC construct diameter compared to each following stage, ****$P<0.001$. Scale bars: 500 µm. (D) 3D-hASC construct total cell number was counted at Day 0 (*n*=9), Day 7 (*n*=9), Day 14 (*n*=9) and Day 21 (*n*=9). Each colour represents a set of experiments. Three independent experiments were performed with *n*=3 biological replicates for each experiment. The *P*-values were obtained using the Mann–Whitney test. Asterisks * indicate the *P*-values of 3D-hASC construct cell number compared to each following stage, **$P<0.01$, ****$P<0.0001$. (E) Longitudinal sections of 3D-hASC constructs were performed at Day 4 and stained with DAPI/Phalloidin to visualise cellular and nuclear shape. Scale bars: 50 µm. (F) Longitudinal sections of 3D-hASC constructs were performed on Day 0, Day 2, Day 4, Day 7, Day 14 and Day 21. The angles of 15 cells were measured on a minimum of three constructs per time point and compared to the axis of tension. Each dot represents the mean of 15 values per 3D-hASC construct. Asterisks * indicate the *P*-values of 3D-hASC construct cell number compared to each following stage, *$P<0.05$, **$P<0.01$. (G) After 0, 2, 4, 7, 14 and 21 Days of culture, 3D-hASC constructs were used for mRNA purification. mRNA expression levels of *COL1A1* were analysed by RT-qPCR. (H) 3D-hASC constructs at Day 4 were transversally cryo-sectioned. 12 µm sections were hybridised with the DIG-labelled antisense probes for *COL1A1*. Scale bar: 250 µm. (I) 3D-hASC constructs at Day 4 Day 0, Day 7, Day 14 and Day 21 were transversally cryo-sectioned. 12 µm sections were hybridised with the DIG-labelled antisense probes for *COL1A1*. Scale bars: 50 µm.

gene expression at the transcript levels with RT-qPCR and *in situ* hybridisation experiments. We first analysed recognised tendon markers at different time points during the culture. The mRNA expression of the following genes was first analysed: the recognised tendon markers *SCX*, *MKX*, *TNMD* and *THSB2*. *SCX* expression was detected, although at low level, both by *in situ* hybridisation and RT-qPCR (Fig. 2A-B). *SCX* expression slightly increased up to Day 7, remained constant up to Day 14 and then decreased up to Day 21 (Fig. 2A). *MKX* expression was not detected at Day 0 and Day 2, but started to be expressed on Day 4 and increased up to Day 21 (Fig. 2C). Interestingly, *TNMD* was expressed in all 3D-hASC constructs on Day 4 but was not detected or at low levels at the other timepoints (Fig. 2D). *THBS2* mRNA expression was significantly increased during human 3D-hASC construct development (Fig. 2E). We also analysed the expression of other genes identified in the top 100 genes (ordered with the highest enrichment score) of *SCX*-positive cells during mouse limb development (Havis et al., 2014), such as *TM4SF1* (Transmembrane 4 superfamily member 1, ranked 4th), *ANXA1* (Annexin A1, ranked 19th) and *S100A10* (member of the S100 family of cytosolic $Ca^{2+}$-binding proteins, ranked 73rd).

*TM4FS1* expression was detected from Day 0 and increased significantly in 3D-hASC constructs over time (Fig. 2G). In contrast, *ANXA1* expression was first decreased at Day 2 and slightly increased at Day 21 (Fig. 2H). The *S100A10* expression levels were stable in 3D-hASC constructs until Day 14 and slightly decreased between Day 14 and Day 21 (Fig. 2I). To address if hASCs lost their progenitor identity, we analysed the expression of recognised molecular markers of mesenchymal progenitors, *HIC1*, *PDGFRA* and *PRRX1* (Arostegui et al., 2022; Lee et al., 2022; Coren et al., 2024) in 3D-hASC constructs. We observed that the three markers were expressed in 3D-hASC constructs at Day 0 (Fig. 2J-L). However, the expression of these markers did not decrease but showed a relatively stable expression over time with sporadic increase of expression (Fig. 2J-L), *HIC1* peaking at Day 4 (Fig. 2J), *PDGFRA* increasing at Day 14 (Fig. 2K) and *PRRX1* slightly increasing at Day 21 (Fig. 2L). Consistent with the activation of the tendon program, 3D-hASC constructs did not express osteogenic or adipogenic markers and did not contain any adipocytes or osteocytes (Fig. S5B-D). It has to be noted that hASCs displayed the ability to differentiate into adipocytes and osteocytes in appropriate culture

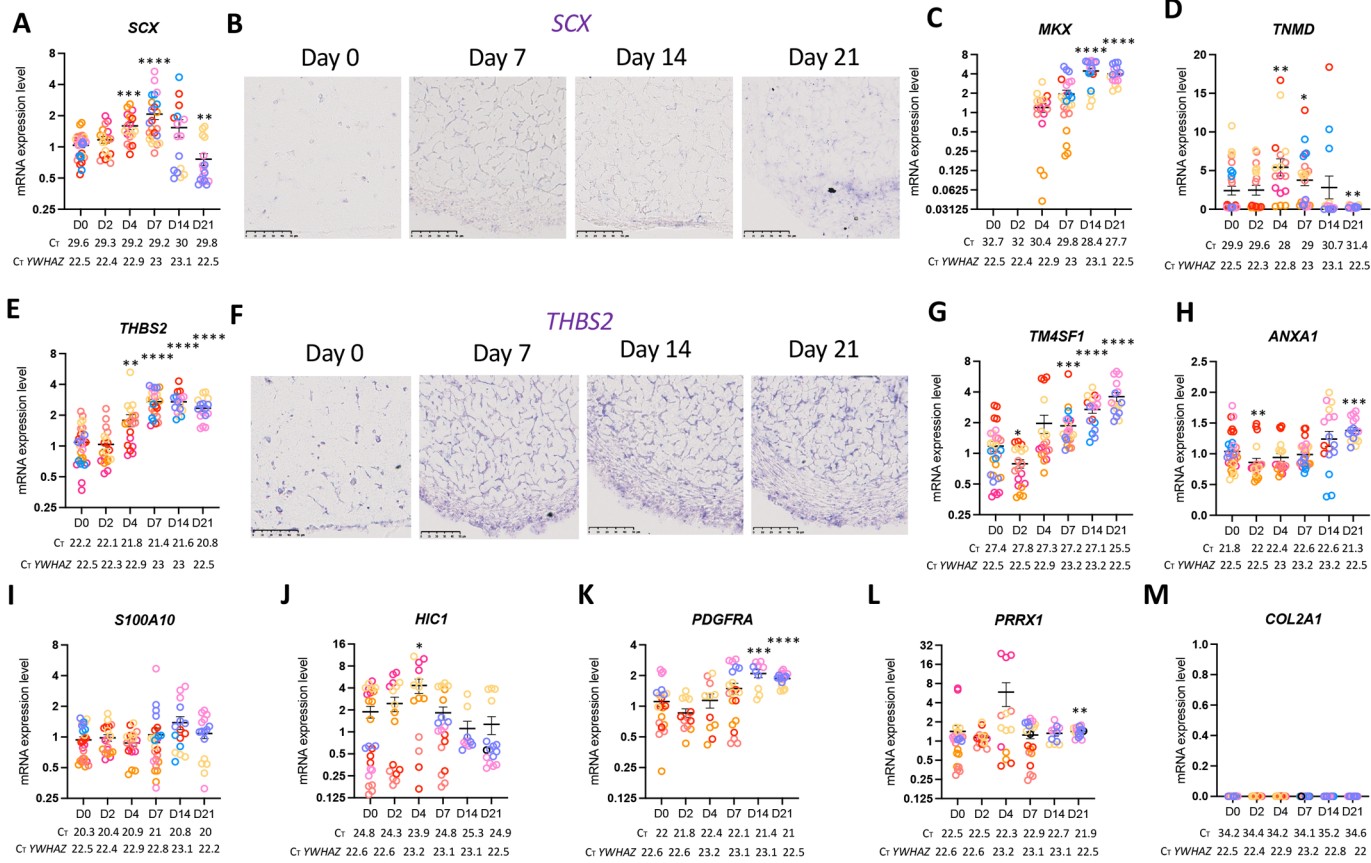

**Fig. 2. 3D-hASC constructs express tendon genes.** (A) After 0, 2, 4, 7, 14 and 21 Days of culture, 3D-hASC constructs were used for mRNA purification. mRNA expression levels of *SCX* were analysed by RT-qPCR. (B) 3D-hASC constructs at Day 0, Day 7, Day 14 and Day 21 were transversally cryo-sectioned. 12 µm sections were hybridised with the DIG-labelled antisense probes for *SCX*. Scale bars: 50 µm. After 0, 2, 4, 7, 14 and 21 Days of culture, 3D-hASC constructs were used for mRNA purification. mRNA samples were analysed by RT-qPCR for (C) *MKX* (D) *TNMD* and (E) *THBS2*. (F) 3D-hASC constructs at Day 0, Day 7, Day 14 and Day 21 were transversally cryo-sectioned. 12 µm sections were hybridised with the DIG-labelled antisense probes for *THBS2*. Scale bars: 50 µm. mRNA expression levels of (G) *TM4FS1*, (H) *ANXA1*, (I) *S100A10*, (J) *HIC1*, (K) *PDGFRA*, (L) *PRRX1*, (M) *COL2A1* were analysed by RT-qPCR. Transcripts are shown relative to the level of *YWHAZ* transcripts. The relative mRNA levels were calculated using the $2^{-\Delta\Delta Ct}$ method, with control being normalised to 1. Each colour represents a set of experiments. Eight independent experiments were performed with 14<*n*<29 3D-hASC construct generated for each step (*n*=29 at Day 0, *n*=19 at Day 2, *n*=18 at Day 4, *n*=25 at Day 7, *n*=17 at Day 14 and *n*=16 at Day 21). The $C_T$ values shown for tendon genes and *YWHAZ* represent the average of all $C_T$ values obtained for each time point. Error bars represent the mean+standard deviations. The *P*-values were obtained using the Mann–Whitney test. Asterisks * indicate the *P*-values of gene expression levels in 3D-hASC constructs compared to the first day of gene detection *$P$<0.05, **$P$<0.01, ***$P$<0.001, ****$P$<0.0001.

media in 2D-culture conditions (Fig. S5A-C) (Kokai et al., 2014; Waldner et al., 2018). The cartilage-specific collagen gene *COL2A1* was not detected 3D-hASC constructs (Fig. 2M), indicating that hASCs had not differentiated into cartilage cells.

We conclude that hASCs undergo towards a tendon program when cultured in 3D-conditions in a collagen gel maintained between two anchor points.

## The Young's modulus of 3D-hASC constructs increases over time

In order to assess whether the activation of the tendon program in hASCs was correlated with a stiffening of the 3D-constructs, we measured the Young's modulus of 3D-hASC constructs in the course of the 21 days of culture. We used nanoindentation (Van Hoorn et al., 2016) to probe the mechanical properties of the 3D-hASC constructs at the local scale (micron scale), with the application of controlled deformations on the surface of the sample. To maintain the 3D-hASC constructs in an immobile position during the nanoindentation experiments, they were detached from their anchor points and deposited onto a support composed of grooves whose length and diameters were adjusted according to constructs geometry. The support was maintained in a Petri dish with agarose gel, while 3D-hASC constructs were fixed to the support using dissection pins inserted through the anchor stems and in the agarose gel (Fig. 3A). Indentation was performed by moving the base of the cantilever at a constant speed of $10 \ \mu m.s^{-1}$, with beads of diameters ranged from 50 to 100 μm (Fig. 3B-D). The relationship between applied force and deformation provides information about 3D-hASC construct mechanical properties. In particular, tissue effective Young's modulus $E_{eff}$ was calculated by fitting the force-indentation curve over a one-micron indentation range using the Hertz model (Fig. 3E, see Materials and Methods). The effective Young's modulus increased significantly during the 21 Days of construct culture starting from 1.3 kPa at Day 0 to 9.2 kPa at Day 21 (Fig. 3F). Notably, $E_{eff}$ increased by twofold factor every 7 days. The effective Young's modulus of 3D-constructs not seeded with cells remains constant for 21 days, at 0.4 kPa (Fig. 3G).

We conclude that 3D-hASC constructs display an increased stiffness over time.

## Extracellular matrix gene expression is correlated with changes in viscoelastic properties of 3D-hASC constructs

Tendon stiffness relies on the synthesis of a dense and organised extracellular matrix generated by tendon cells. We thus analysed the expression of transcripts encoding tendon-associated collagens and fibrillogenesis-associated proteins during 3D-hASC constructs formation. Similarly to *COL1A1* (Fig. 1G), the mRNA levels of the tendon-associated collagens *COL6A3* and *COL14A1* were detected at Day 0, increased until Day 7 and remained higher at Day 21 compared to Day 0 (Fig. 4A-B). We also analysed the expression of genes encoding fibrillogenesis-associated proteins, *DPT*, *DCN*, *POSTN* and *LOX*, which exhibit an elevated enrichment score in E14.5 compared to E11.5 *SCX*-positive cells during mouse limb development (Havis et al., 2014). The transcript levels of *DPT*, *DCN*, *POSTN* and *LOX* were significantly increased during human 3D-hASC construct formation (Fig. 4C-F). The persistent increase of tendon collagen and collagen fibrillogenesis gene expression was indicative of matrix production and parallel the increase of effective Young's modulus (Fig. 3F) during 3D-hASC construct formation. In addition, the increased expression of *DPT*, *DCN*, *POSTN*, *THBS2* and *LOX*, regulators of ECM network, also suggested that the

structure and organisation of the ECM was modified during the course of 3D-hASC construct formation. We thus tested whether ECM transcript expression was accompanied with a modification of the viscoelastic properties of 3D-hASC constructs along time. To do this, we carried out stress relaxation experiments using nanoindentation (Fig. 5). In particular, we calculated the percentage of relaxation $r_{relax}$ from the peak force $F_{max}$ measured at the end of the indentation ramp and the force $F_{t=10s}$ measured after 10 s of relaxation [$r_{relax}=(F_{max}-F_{t=10s})/F_{max}$]. The higher this ratio is, the more liquid-like the material. Oppositely, a smaller ratio indicates more solid, elastic-like viscoelastic properties. On Day 0, the percentage of relaxation of 3D-hASCs was 35% and decreased along time down to 24% on Day 21 (Fig. 5E), in parallel with the increase of ECM transcript levels. Interestingly, the percentage of relaxation of 3D-no cell constructs on Day 0 was 23%, decreased to 15% on Day 7 and remained constant (Fig. 5E).

We conclude that the 3D-hASC constructs acquire solid-like properties, consistent with the expression of matrix transcripts during the time course of the cultures.

## Cells in 3D-hASC constructs self-organise into two cell populations

In order to analyse cell organisation, we performed longitudinal and transverse sections of 3D-hASC constructs (Fig. 1E, Fig. 6; Fig. S6). We observed that cells in the 3D-hASC constructs organise into a central region surrounded by a dense surface layer as early as Day 2 (Fig. 6B). Phalloidin staining showed that the surface layer was three to four cells deep from Day 4 to Day 21, and was separated from the central region by a sharp interface (Fig. 6B). The cells at the periphery of the 3D constructs showed a significantly higher cell density than the centre (Fig. 6C). However, cell density remained constant over time in both the central and peripheral layers (Fig. 6C). Interestingly, ECM gene expression evolved differently in the two cell layers from Day 4 (Figs 1I and 2F). *COL1A1* was expressed in both central and peripheral cells of 3D-hASC constructs, as observed by *in situ* hybridisation from Day 7 to Day 14 (Fig. 1I). However, at Day 21, *COL1A1* showed stronger expression in central cells compared to peripheral cells (Fig. 1I). *THBS2* was visualised in both central and peripheral cells throughout the 21 days of culture (Fig. 2F). Finally, a marker of peripheral cells of adult rat tendon, *COL3A1* (Steffen et al., 2023) was detected both in both central and peripheral cells (Fig. S7).

We conclude that cells within 3D-hASC constructs self-organise into two populations, one in the centre and one at the periphery.

## DISCUSSION

This study showed that hASCs enter the tendon differentiation program when cultured in a minimal 3D-system. hASCs embedded into a cylindrical scaffold of type I-collagen attached to two anchor points acquired stiffness and solid-like properties that parallel the expression of tendon matrix genes.

## The genetic program activated in 3D-hASCs mimics that of tendon development

With no external molecular cues other than the culture medium, hASCs mixed in a 3D-collagen gel expressed tendon genes encoding transcription factors regulating *COL1A1* transcription and tendon matrix genes. The expression of the developmental tendon genes, *SCX*, *MKX* and *TNMD* in 3D-hASCs is reminiscent of their molecular progression during development: *SCX* and *MKX* are sequentially expressed in 3D-hASC constructs, as during mouse tendon development (Ito et al., 2010; Liu et al., 2010). The expression pattern of *SCX* displays a bell shape, while the tendon matrix gene

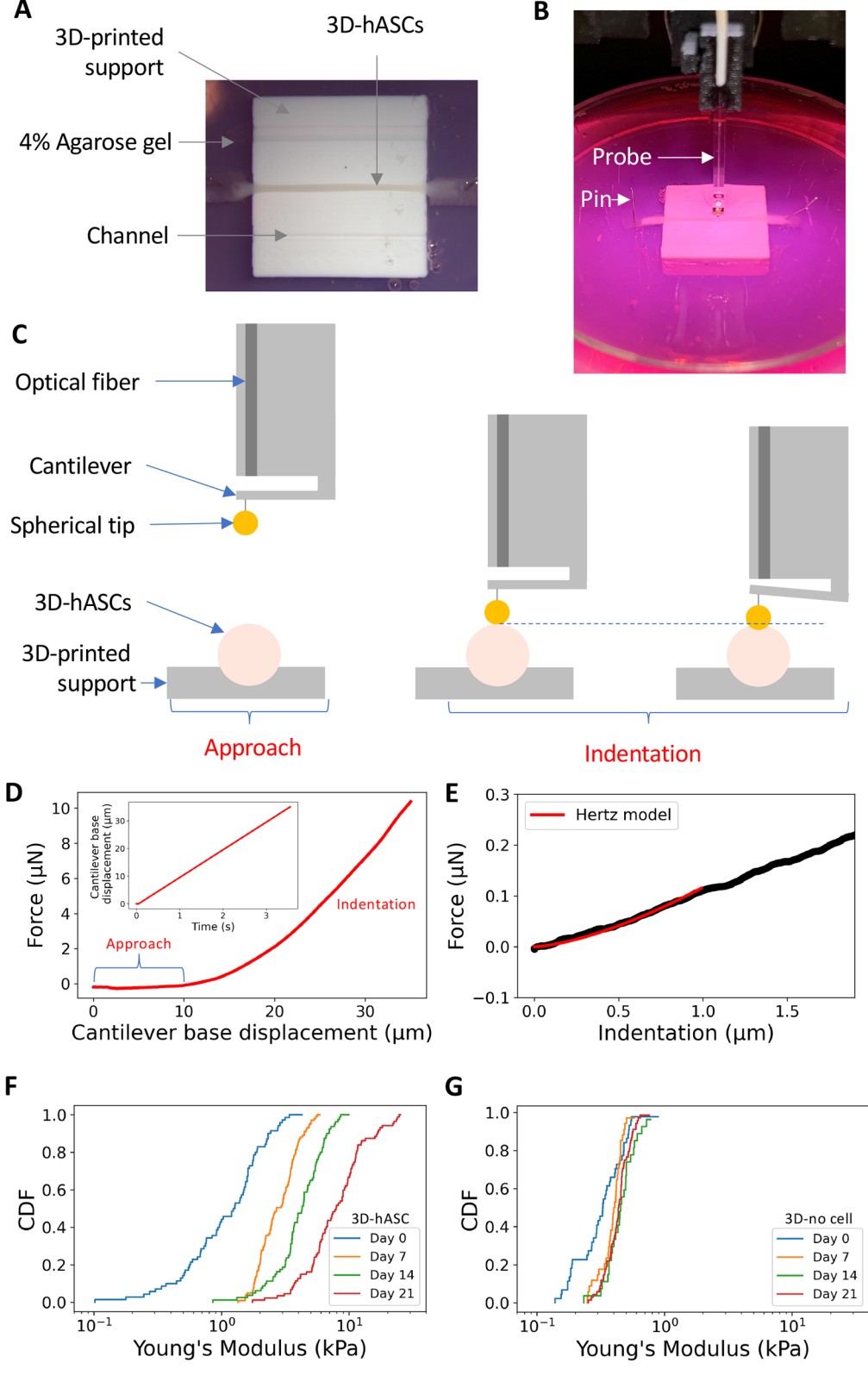

**Fig. 3. 3D-hASC construct Young's modulus increased along time.**
(A) 3D-hASC constructs are fixed on a 3D-printed support maintained in agarose gel. (B) 3D-hASC constructs are placed in an appropriate channel according to their diameter and fixed using dissection pins inserted into the anchor stems and agarose. (C) The one-piece optical probe of the nanoindenter is composed of an optical fibre, a cantilever and a spherical tip. After an approach phase, the tip touches the sample and indent it. The cantilever bending is detected by the optical fibre. (D) Force as a function of cantilever base displacement for a typical indentation. Inset: cantilever base displacement is applied at constant rate of 10 µm.s-1. (E) Force as a function of indentation for a typical indentation curve. Hertz contact fit (red) is applied over a 1 µm indentation. (F) Cumulative distribution function (CDF) of effective Young's modulus along culture time in 3D-hASCs. The 0.5 CDF value corresponds to the median Young's modulus value. Indentation experiments were performed at Day 0 ($n$=17), Day 7 ($n$=17), Day 14 ($n$=13) and Day 21 ($n$=14). Five independent experiments were performed with 3<$n$<8 biological replicates for each experiment. (G) Cumulative distribution function (CDF) of effective Young's modulus along culture time in 3D-constructs without cells. Indentation experiments were performed at Day 0 ($n$=8), Day 7 ($n$=6), Day 14 ($n$=6) and Day 21 ($n$=12). Two independent experiments were performed with 3<$n$<6 biological replicates for each experiment.

expression increases over time in 3D-hASC constructs, consistent with SCX function as a starter of the developmental tendon program. The sequential expression of *SCX* and *TNMD* and the decrease of *SCX* and *TNMD* at Day 21 in the 3D-hASC constructs are consistent with *TNMD* regulation by SCX (Murchison et al., 2007; Shukunami et al., 2018). *TM4SF1* and *ANXA1,* coding for a Tetraspanin transmembrane protein and AnnexinA1, respectively, were identified in the top 100 genes (ordered with the highest enrichment score) of *SCX*-positive cells during mouse limb development (Havis et al., 2014). They display a regular increase over time as the *LOX* gene (lysyl oxidase proteins). Interestingly, the *TM4SF1*, *ANXA1* and *LOX* genes have been identified in a fibroblast cluster for good prognosis in

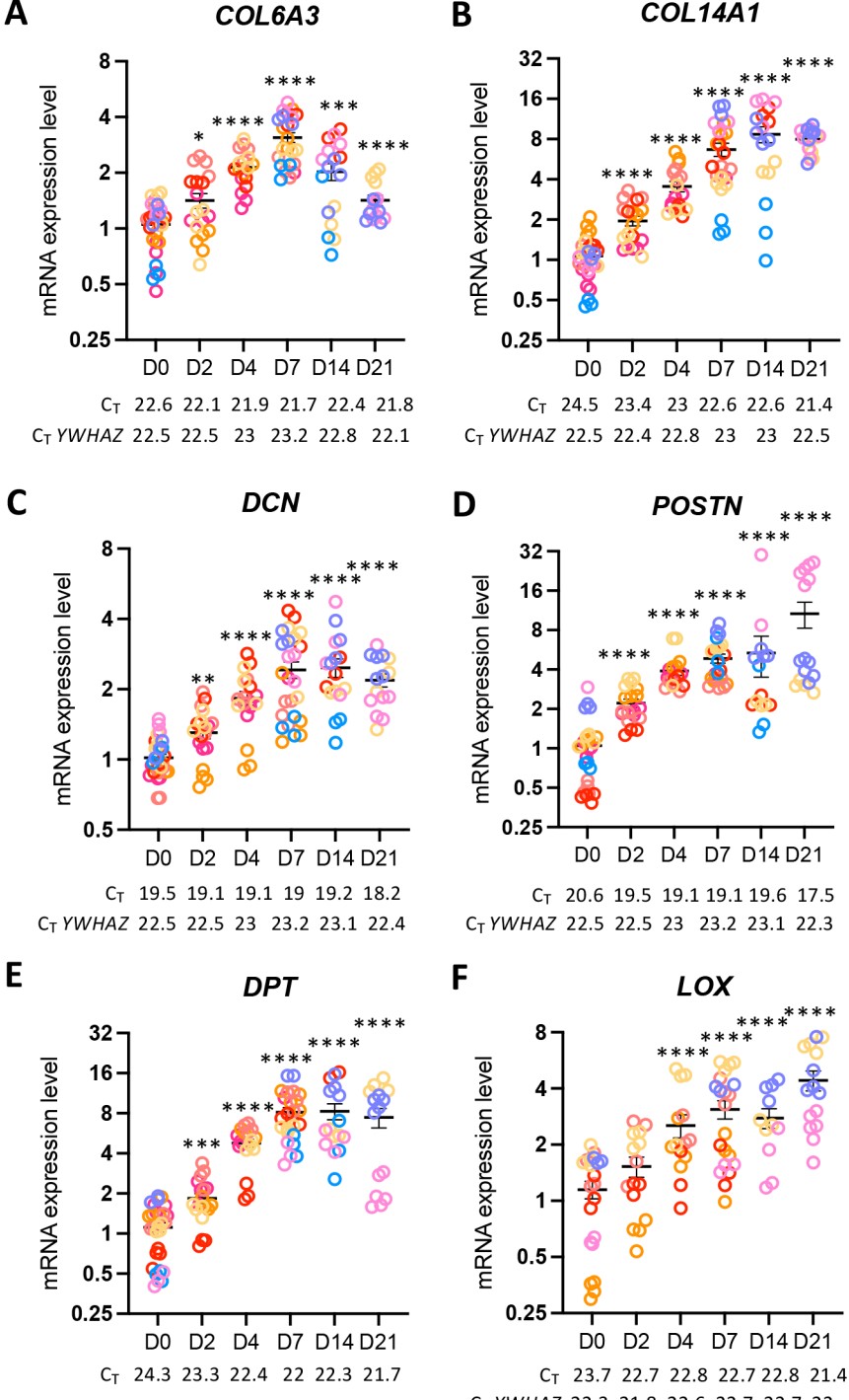

**Fig. 4. Extra-cellular matrix gene expression was highly increased during 3D-hASC construct development.** After 0, 2, 4, 7, 14 and 21 Days of culture, 3D-hASC constructs were used for mRNA purification. mRNA expression levels of tendon-associated collagens (A) *COL6A3*, (B) *COL14A1* and extracellular matrix protein coding genes (C) *DCN*, (D) *POSTN*, (E) *DPT* and (F) *LOX* were analysed by RT-qPCR. Transcripts are shown relative to the level of *YWHAZ* transcripts. The relative mRNA levels were calculated using the $2^{-\Delta\Delta Ct}$ method, with control being normalised to 1. Each colour represents a set of experiments. Eight independent experiments were performed with 14<*n*<29 3D-hASC construct generated for each step (*n*=29 at Day 0, *n*=19 at Day 2, *n*=18 at Day 4, *n*=25 at Day 7, *n*=17 at Day 14 and *n*=16 at Day 21). The $C_T$ values shown for tendon genes and *YWHAZ* represent the average of all $C_T$ values obtained for each time point. Error bars represent the mean + standard deviations. The *P*-values were obtained using the Mann–Whitney test. Asterisks * indicate the *P*-values of gene expression levels in 3D-hASC constructs compared to the first day of gene detection **P*<0.05, ***P*<0.01, ****P*<0.001, *****P*<0.0001.

bladder cancer (Wang et al., 2024) and are seen as therapeutic targets for fibrosis (Chen et al., 2019; Yan et al., 2022; Zhao et al., 2024). ANXA1 and S100A10 belong to two families of proteins known to form hetero-tetramers (Réty et al., 2000), suggesting that ANXA1 interacts with S100A10 to maintain the tendon phenotype of 3D-hASC constructs.

The mRNA expression levels of the tendon-associated collagen genes (*COL1A1*, *COL6A3*, *COL14A1*) and genes encoding proteins involved in tendon collagen fibrillogenesis (*THBS2*, *DCN*, *POSTN*, *DPT*) rise sharply during the formation of 3D-hASC constructs, and were maintained at high levels at the end of culture. These results indicate that the reduced expression of *SCX* in 3D-hASCs does not affect the molecular identity of tendon ECM in the constructs. The expression of the mesenchymal stem cell markers (*HIC1*, *PDGFRA* and *PRRX1*) did not decrease in 3D-hASCs over time, which is consistent with their expression in the tendon lineage (Harvey et al., 2019; Arostegui et al., 2023; Hirsinger et al., 2024). The peak of *HIC1* expression at Day 4 suggests that the hASCs are undergoing a teno-chondrogenic progenitor step (Arostegui et al., 2023). However, the hASCs in 3D cultures favour a tendon fate, as cartilage markers are not expressed in 3D-hASC constructs.

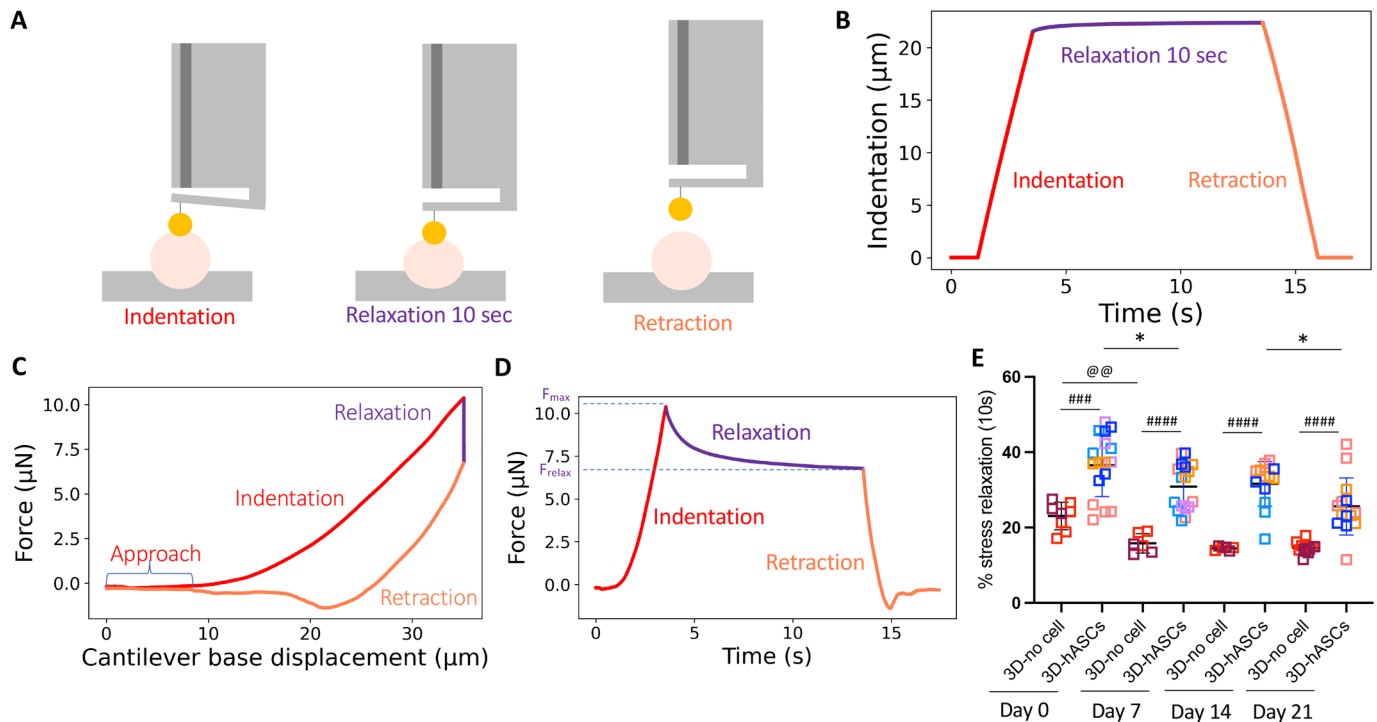

**Fig. 5. Viscoelastic properties of human 3D-hASC constructs change over the course of the culture.** (A) Schematic of a force relaxation experiment. In open loop, the base of the cantilever is fixed while relaxation takes place. (B) Typical curve showing force as a function of indentation during loading, relaxation and unloading phase. Inset: indentation as a function of time. (C) Force as a function of cantilever base displacement for a typical indentation. Inset: cantilever base displacement is applied at constant rate of 10 µm s$^{-1}$. (D) Force as a function time. Force peaks at a value Fmax and reaches is close to a plateau after 10 s of relaxation. Frelax is defined as the force at this instant. (E) Percentage of stress relaxation [(Fmax−Ft=10 s)/Fmax)] ×100 during the first 10 s of relaxation, after different periods of cell culture. For 3D-no cell constructs without cells, each dot represents the mean of five measurements performed on one construct. Indentation experiments were performed at Day 0 ($n$=8), Day 7 ($n$=6), Day 14 ($n$=6) and Day 21 ($n$=12). Each colour represents a set of experiments. Two independent experiments were performed with 3<$n$<6 biological replicates for each experiment. For 3D-hASC constructs, indentation experiments were performed at Day 0 ($n$=17), Day 7 ($n$=17), Day 14 ($n$=13) and Day 21 ($n$=14). Each colour represents a set of experiments. Five independent experiments were performed with 3<$n$<8 biological replicates for each experiment. The $P$-values were obtained using the Mann–Whitney test. Asterisks * indicates the $P$-value of 3D-hASC construct relaxation compared to each following stage; *$P$<0.05; @ indicates the $P$-value of 3D-no cell construct relaxation compared to each following stage, @$P$<0.05; # indicates the $P$-value of in 3D-no cell constructs compared to 3D-hASC constructs, ###$P$<0.001, ####$P$<0.0001.

## Progression of mechanical properties of hASCs in 3D-cultures

During the 21-day culture period, the 3D-hASCs showed reduced diameter, increased stiffness and reduced relaxation. The reduction in the diameter of 3D-hASC constructs was already observed in 3D-constructs using human dermal fibroblasts (HDF01035) or calf patellar tendon cells embedded in a collagen gel (Schulz Torres et al., 2000; Vaughan et al., 2019). Because the diameters of 3D-constructs without any cell remained constant, we conclude that hASCs were responsible for the collagen compaction of 3D-hASC constructs.

There is a linear increase of the stiffness of 3D-hASC constructs during the time of culture. Notably, E$_{eff}$ increased by two-fold factor every 7 days. This indicates that this long time-scale stiffening is unlikely to be due to ECM compaction, which occurred almost exclusively during the first week of culture. We rather believe that the expression of genes encoding specific ECM components would drive collagen compaction. After 3 weeks of cell culture, 3D-hASC constructs exhibited a Young's modulus values comprised between 5.6 and 18.3 kPa. Interestingly, measurements of nanoscale elastic modulus of chicken foetal calcaneus tendons calculated similarly through Hertz contact theory gave results of same order of magnitude (5-17 kPa) (Marturano et al., 2013; Connizzo and Grodzinsky, 2017). This suggests that 3D-hASC constructs mimic

the progression of mechanical properties observed during tendon development. In studies using human adult tendon fibroblasts (not MSCs) to generate 3D constructs, Young's modulus measurements lead to a value close to that found by tensile tests in adult tendons, of the order of MPa (Herchenhan et al., 2013; Avey et al., 2024).

## Link between matrix gene expression and mechanics in 3D-hASC constructs

The increase of expression of ECM-encoding genes parallels the stiffness increase and concomitant relaxation reduction of 3D-hASC constructs. We observed a temporal correlation between ECM gene expression changes and increased stiffness in the 3D constructs. We see this correlation as a bidirectional interaction between mechanical properties and matrix components during the time of the culture. It is recognised that changes of physical properties act on matrix composition and conversely, matrix interferes with mechanical properties (Saraswathibhatla et al., 2023). We found that hASCs progressively acquired a spindle-shaped form associated with elongation of the cell shape during the course of the 3D-culture, as observed in native tendons. The elongated shape of cells observed in native tendons and 3D-hASC constructs is probably involved in tendon matrix production, since nuclear response upon deformation is recognised to include changes in transcriptional activity (Miroshnikova and Wickström, 2022).

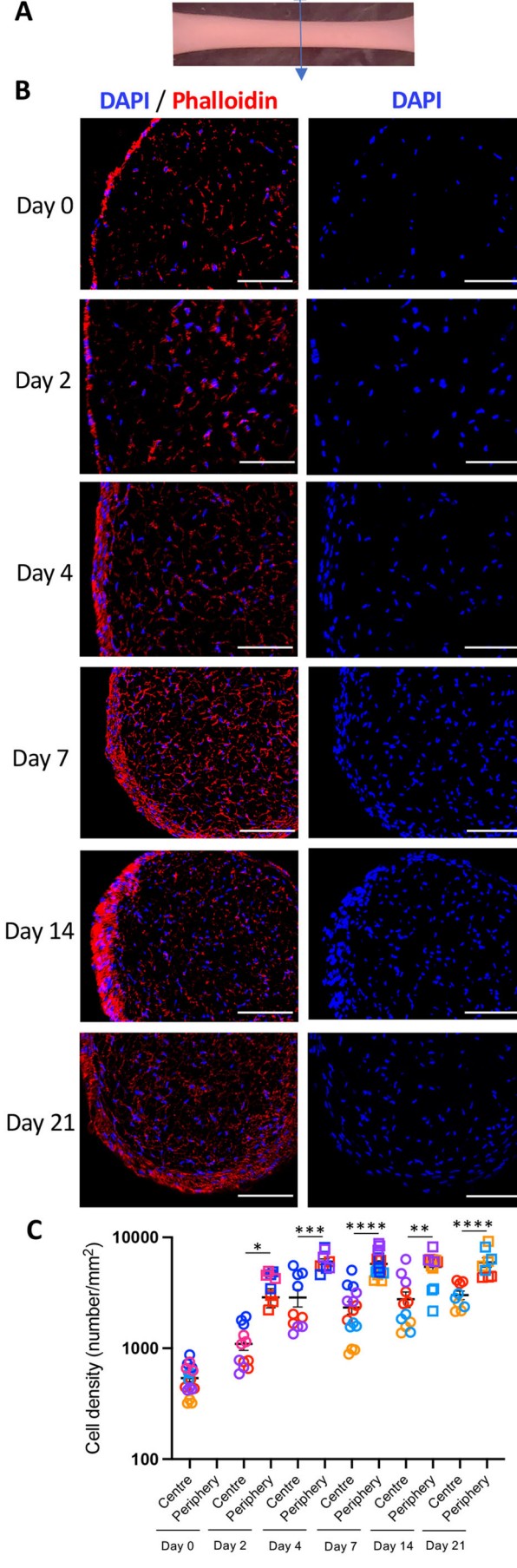

**Fig. 6. Cellular organisation in 3D engineered 3D-hASCs.** (A) 3D-hASCs transverse section representation. (B) Transverse sections of 3D-hASC constructs were performed on Day 0, Day 2, Day 4, Day 7, Day 14 and Day 21 of culture and stained with DAPI/Phalloidin to visualise cell organisation within the 3D-hASC constructs. From Day 2, 3D-hASC constructs are composed of two cell populations, one in the centre and the other at the periphery. (C) Cell density (cell number per $mm^2$) was counted at the centre and periphery of 3D-hASC constructs at Day 0 ($n$=18), Day 2 ($n$=12), Day 4 ($n$=10), Day 7 ($n$=15), Day 14 ($n$=12) and Day 21 ($n$=9). Each colour represents a set of experiments. Six independent experiments were performed with 3<$n$<4 biological replicates for each experiment. The $P$-values were obtained using the Mann–Whitney test. Asterisks * indicate the $P$-values of cell density at the centre compared to the ring of 3D-hASCs. *$P$<0.05, **$P$<0.01, ***$P$<0.001, ****$P$<0.0001. Scale bars: 100 μm.

Cells organise in space within 3D-hASCs into distinct central and peripheral layers. Such organisation has been previously described in 3D-constructs generated with the Flexcell technology either with hASCs (Yang et al., 2013) or avian tendon internal fibroblasts (Garvin et al., 2003). This cell organisation may be an effect of the geometry of the constructions defined by the anchoring points, which may orient cell shape and ECM. The peripheral cell layer of 3D-hASC constructs is reminiscent of the peritenon external layer surrounding native tendon. This is consistent with the lower *COL1A1* expression in the peripheral cells compared to central cells in 3D-hASC constructs at Day 21 (Fig. 1I), also observed in equine and rat peritenon cells versus tendon core cells (Pechanec et al., 2020; Steffen et al., 2023). *COL3A1* was identified recently as a peritenon marker in adult rat patellar tendons (Steffen et al., 2023). However, *COL3A1* was detected both at the centre and the periphery of 3D-hASC constructs (Fig. S7) indicating that *COL31A1* expression does not discriminate the peritenon identity between peripheral and central cells in 3D-hASC constructs. Interestingly, a peripheral cell layer has been identified surrounding developing tendons in mouse and chicken embryos (Eloy-Trinquet et al., 2009; Havis et al., 2014; Hirsinger et al., 2024), suggesting that future peritenon cells are present early during the tendon program.

Although the 3D-hASC constructs display a tendon matrisome gene expression profile at 3 weeks, we do not exclude that external mechanical cues would improve the tendon phenotype of 3D-hASC constructs as already reported (Wang et al., 2018; Park et al., 2022). During development, muscle contractions are recognised to maintain *SCX* expression and the tendon phenotype in developing tendons (Gaut and Duprez, 2016). An interesting hypothesis is that application of external mechanical cues would prevent *SCX* and *TNDM* downregulation and allow *HIC1* downregulation in 21-day-old 3D-hASC constructs.

In summary, we have established that hASCs cultured in a minimal 3D-system progress into the tendon differentiation program along 3 weeks of culture. The molecular tendon program is associated with variations of mechanical properties reminiscent of embryonic tendon development.

## MATERIALS AND METHODS
### Cell sample collection
Human abdominal subcutaneous adipose tissues were obtained from healthy donors after liposuction surgery (AP-HP Saint-Antoine Hospital, Paris, France). Human Adipose stromal cells (hASCs) were isolated from three healthy donors (one female and two male).

All donors provided their written, informed consent to the use of their tissue specimens for research purposes. The study was performed in compliance with the principles of the Declaration of Helsinki and was approved by an institutional review board. The study was approved by the French regulatory authorities (Codecoh DC2023-5617). Cells were

cryopreserved in 2 ml cryotubes (2×106 cells per vial) stored in liquid nitrogen tanks.

## Cell amplification

For each experiment, cell vials were thawed 2-3 min in a 37°C water bath and transferred in T25 cell culture flasks (Gibco) with 5 ml of hASCs culture medium containing minimum essential medium (αMEM; Gibco; ref 22571-020) complemented with 10% fetal calf serum (FCS; Gibco; ref 35-079-CV), 1% HEPES (1 M, sigma; ref H0887-100 ml), 1% glutamine (100X, 200 mM, Gibco; ref 25030-024), 1% penicillin/streptomycin (Gibco; ref 15140-122) and 2.5 ng/ml FGF2 (PeproTech, Rocky Hill, NJ, USA; ref 100-18B). Confluent cells were submitted to two passages, at 1:9 and 1:3 ratio. For each passage, we used 0.05% Trypsin (Gibco; ref 25300-054) for 5 min.

## Preparation of 3D-hASC constructs

3D-hASC constructs were prepared as described (Scott et al., 2011). Briefly, sub confluent cells were detached in 5 ml of 0.05% trypsin for 5 min, then suspended with hASC culture medium (described below) and counted with a Malassez counting chamber. 16.2 million cells (600,000 cells per 3D-hASC construct) were centrifuged at 1200 rpm for 5 min and resuspended in 4.05 ml of Type I collagen solution containing 70% Purecoll (Sigma-Aldritch, Oakville, Canada; ref 5074-35ml), 10% FCS and 20% αMEM brought to 7.4 pH with 0.1 N NaOH immediately before use. 150 µl of hASCs-collagen mix was pipetted into each well of an untreated TissueTrain© plate (FlexCell International, Hillsborough NC, USA; ref TTLC-4001C) according to the manufacturer's instructions. The collagen hydrogel was allowed to set for 2 h, and the 3D-hASC constructs were covered with 2 ml of hASC culture medium completed with 0.25 mM ascorbic acid. The medium was replaced every 2-3 days.

## 3D-hASC construct sections

3D-hASC constructs were fixed in 1% paraformaldehyde (PFA, Gibco Sigma; ref 158127-100G) overnight at 4°C, washed in 1× PBS (Gibco; ref 14190-094) and put in a 15% sucrose solution at 4°C for 3 days. Before inclusion, 3D-hASC constructs were incubated in a 7.5% gelatin–15% sucrose solution in a 37°C water bath for 1 h. Included 3D-hASC constructs were then frozen at −80°C in isopentane for 1 min and 12 µm longitudinal and transversal cryosections were generated with a CM3050 S Cryostat (Leica Biosystems).

## Cell angle measurements

Longitudinal sections were stained with DAPI (D9542, Sigma-Aldrich) to visualise cell nuclei and phalloidin (A12380 Alexa Fluor Phalloidin, Invitrogen/ ThermoFisher scientific) to visualise cytoskeletal F-actin. Fluorescent images were captured using a Zeiss Axio Observer Z1 microscope equipped with a Zeiss Apotome 2 and an Axiocam 506 monochrome camera. Cell/nucleus angles relative to the axis of the 3D-hASC constructs were measured using Fiji Image J software.

## RNA isolation

Entire 3D-hASC constructs were put in 500 µl RLT (Qiagen; ref 74106) and homogenised using a mechanical disruption device (Lysing Matrix A, Fast Prep MP1, 4×30 s, 6 m s$^{-1}$). Total RNA was isolated using the RNeasy mini kit (Qiagen; ref 74106), according to the manufacturer's instructions, including a 15 min of DNase I (Qiagen; ref 79254) treatment.

## Reverse transcription and quantitative real time PCR

500 ng of total RNAs extracted from each 3D-hASC construct, for each timepoint, were reverse transcribed using the high-capacity retro-transcription kit (Applied Biosystems; ref 4387406). Quantitative PCR analyses were performed using primers listed in Table S1 and SYBR Green PCR Master Mix (Applied Biosystems; ref 4385614). The relative mRNA levels were calculated using the $2^{-\Delta\Delta Ct}$ method (Livak and Schmittgen, 2001). The cycle threshold ($C_T$) corresponds to the number of cycles required for the fluorescent signal to cross the threshold (to exceed background level). The $\Delta C_T$s were obtained from $C_T$ normalised to YWHAZ $C_T$ level in each sample. The primers used for RT-qPCR experiments are listed in Table S1. Most primers were designed in the laboratory, except for hMKX (Bayer et al.,

2014), hPPARG (Waldner et al., 2018), hS100A10 (Głowacka et al., 2021) , hTHBS2 (Xu et al., 2020), hYWHAZ (Ragni et al., 2013).

## In situ hybridisation

3D-hASC construct 12 µm cryosections were subject to a standard in situ hybridisation protocol (Bonnin et al., 2005). COL1A1, SCX, THBS2 and COL3A1 and probes were generated using a Roboprobe kit (Promega; ref P1460), according to manufacturer's instructions. Images were captured using a Nanozoomer SQ Digital Slide Scanner with NDP.view2 software (Hamamatsu Photonics). The primers used for probe synthesis are listed in Table S1.

## Nanoindentation measurements

Live 3D-hASC constructs were detached from their anchor points and immediately fixed onto a 3D-printed support. To avoid sliding of the construct during mechanical testing, the support was designed with three half-cylindrical grooves at its surface whose diameters were adjusted to the typical 3D-hASC constructs diameters. To conserve the original length of the 3D-hASC construct, the length of the support was also calibrated to the original distance between the anchor points in the bioreactor. The support was then embedded in an agarose gel in a Petri dish, and the samples were immersed in culture medium. Mechanical testing was performed at room temperature, using a Piuma nano-indenter (Optics11 Life), as described in the main text. Samples were positioned under the probe using a magnifying glass. Five measurements were performed along the 3D-hASC construct length and their mean value was calculated to extract effective Young's modulus and percentage of stress relaxation for each sample.

The effective Young's modulus $E_{eff}$ was calculated from a fit of the force versus indentation curve over a 1 µm indentation course and using a Hertzian contact model as follows:

$$F = \frac{4}{3} E_{eff} \sqrt{R_i} \cdot h^{3/2},$$

where F is the force, Ri the tip radius, and h the indentation.

## Statistical analyses

Data were analysed using the non-parametric Mann–Whitney test with Graphpad Prism version 10. Results are shown as means±standard deviations. The P-values are indicated either with the value or with * or #. Values more than 2SD above or below the mean were excluded.

## Cleaved caspase 3 and phospho-histone H3 immunostaining

3D-hASC construct 12 µm cryosections were subject to a standard immunofluorescence protocol using cleaved Caspase-3 (Asp175) (5A1E) rabbit mAb (Cell Signaling; ref 9664S) or phospho histone-H3 (S10) rabbit Ab (Cell Signaling; ref 9701S) primary antibody at 1:100 dilution and goat anti-rabbit IgG (H+L) cross-adsorbed, Alexa Fluor 488 (Invitrogen/ ThermoFisher Scientific; ref A11008) secondary antibody at 1:200 dilution. Finally, sections were stained with DAPI (D9542, Sigma-Aldrich) to visualise cell nuclei and phalloidin (A12380 Alexa Fluor Phalloidin, Invitrogen/ ThermoFisher scientific) to visualise cytoskeletal F-actin. Fluorescent images were captured using a Zeiss Axio Observer Z1 microscope equipped with a Zeiss Apotome 2 and an Axiocam 506 monochrome camera.

## COL1A1 immunostaining

3D-hASC construct 12 µm cryosections were subject to a standard immunofluorescence protocol using human COL1A1 (Abcam; ref ab260043) primary antibody at 1:100 dilution and goat anti-rabbit IgG (H+L) cross-adsorbed, Alexa Fluor™ 488 (Invitrogen/ThermoFisher scientific; ref A11008) secondary antibody at 1:200 dilution. Finally, sections were stained with DAPI (Sigma-Aldrich; ref D9542) to visualise cell nuclei and phalloidin (Alexa Fluor Phalloidin, Invitrogen/ThermoFisher scientific; ref A12380) to visualise cytoskeletal F-actin. Fluorescent images were captured using a Zeiss Axio Observer Z1 microscope equipped with a Zeiss Apotome 2 and an Axiocam 506 monochrome camera.

## Adipogenic differentiation

hASCs from each patient were plated at passage 3 at a density of 3000 cells/cm$^2$ in six-well plates pre-coated with animal component-free

(ACF) cell attachment substrate (Stemcell Technologies; ref 07130). Cells were cultured in ACF Plus medium until confluence. Control cells were cultured in ACF Plus medium (Stemcell Technologies; ref 05446) whereas adipogenic differentiation was tested by replacing the ACF Plus medium with human Mesencult adipogenic differentiation medium (Stemcell Technologies; ref 05412) for 13 days. Medium was changed every 2-3 days.

## Osteogenic differentiation

hASCs were plated at passage 3 at a density of 8000 cells/cm$^2$ in six-well plates pre-coated with animal component-free (ACF) cell attachment substrate (Stemcell Technologies; ref 07130). Cells were cultured in ACF Plus medium until confluence. Control cells were cultured in ACF Plus medium (Stemcell Technologies; ref 05446) whereas osteogenic differentiation was tested by replacing the ACF Plus medium with human Mesencult osteogenic differentiation medium (Stemcell Technologies; ref 05465) for 14 days. Medium was changed every 4 days.

## Cell staining

Adipogenic differentiation was assessed by Oil Red O staining and osteogenic differentiation was assessed by Alizarin Red Staining, as previously described (Waldner et al., 2018; Bléher et al., 2020).

## RNA isolation

Control and differentiation media were removed and replaced with 350 μl RLT in each well or tube. Total RNA was isolated using the RNeasy mini kit (Qiagen; ref 74106), according to the manufacturer's instructions, including a 15 min of DNase I (Qiagen; ref 79254) treatment.

## Reverse-transcription and quantitative real time PCR

Total RNAs extracted from control, adipogenic or osteogenic cells were reverse transcribed using the high-capacity retro-transcription kit (Applied Biosystems; ref 4387406). Quantitative PCR analyses were performed using primers listed in Table S1 and SYBR Green PCR Master Mix (Applied Biosystems; ref 4385614). The relative mRNA levels were calculated using the $2^{-\Delta\Delta Ct}$ method (Livak and Schmittgen, 2001). The Cts were obtained from Ct normalised to *YWHAZ* level in each sample.

## Acknowledgements
We would like to thank Mathieu Hautefeuille for proofreading the manuscript.

## Competing interests
The authors declare no competing or financial interests.

## Author contributions
Conceptualization: J.F., D.D., E.H.; Data curation: L.G.P.; Formal analysis: J.F.; Funding acquisition: D.D.; Investigation: M.H., L.G.P.; Methodology: M.H., J.F., X.L., C.B., L.G., E.H.; Resources: M.H., V.B., C.L.; Supervision: D.D., E.H.; Validation: E.H.; Writing – original draft: C.L., D.D., E.H.; Writing – review & editing: J.F., V.B., D.D., E.H.

## Funding
This work was supported by the Centre National de la Recherche Scientifique (CNRS), Institut national de la santé et de la recherche médicale (Inserm), Sorbonne Université and Agence Nationale de la Recherche (ANR_TENORS). Open Access funding provided by Sorbonne Université. Deposited in PMC for immediate release.

## Data and resource availability
All relevant data can be found within the article and its supplementary information. All the data and resources generated in this study are available upon request.

## Peer review history
The peer review history is available online at https://journals.biologists.com/bio/article-lookup/doi/10.1242/bio.061911.

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
