## [Peer Review File · Biology Open]

Human adipose stromal cells differentiate towards a tendon phenotype with adapted visco-elastic properties in a 3D-culture system

Maxime Horde, Jonathan Fouchard, Luna Gomez Palacios, Xavier Laffray, Cedrine Blavet, Veronique Bereziat, Claire Lagathu, Ludovic Gaut, Delphine Duprez and Emmanuelle Havis
DOI: 10.1242/bio.061911

Editor: Alissa Armstrong

Review timeline

Original submission:	27 January 2025
Editorial decision:	4 February 2025
First revision received:	8 April 2025
Accepted:	14 April 2025

Original submission

First decision letter

MS ID#: bio.061911

MS TITLE: Human adipose stromal cells differentiate towards a tendon phenotype with adapted visco-elastic properties in a 3D-culture system

AUTHORS: Maxime Horde; Jonathan Fouchard; Luna Gomez Palacios; Xavier Laffray; Cedrine Blavet; Veronique Bereziat; Claire Lagathu; Ludovic Gaut; Delphine Duprez; Emmanuelle Havis

I have now reached a decision on the above manuscript.

The reviewer reports are shown at the bottom of this email or can be accessed, together with a copy of this decision letter, by going to:

As you will see, the reviewers gave favourable reports, but raised some critical points that will require amendments to your manuscript. I hope that you will be able to carry these out, because we would like to be able to accept your paper.

At this stage, we also ask you to ensure your manuscript complies with our formatting guidelines - please see our manuscript preparation guidelines for details. Provided you are able to fully address the referees' comments, we are positive about publication of your paper (we accept over 95% of revision submissions) and therefore hope you won't mind any extra work involved in reformatting your manuscript at this point.

Please ensure that you clearly highlight all changes made in the revised manuscript. Please avoid using 'Tracked changes' in Word files as these are lost in PDF conversion

Reviewer 1

Comments for the author

The manuscript by Hordé et al., titled "Human adipose stromal cells differentiate towards a tendon phenotype with adapted visco-elastic properties in a 3D-culture system", investigates tendon cell differentiation in human adipose stromal cells (hASCs) using a minimal 3D system. By embedding hASCs in a type-I collagen gel with a static uniaxial constraint, the authors evaluated tendon-associated gene expression and mechanical properties over three weeks. The results demonstrate a molecular progression towards a tendon phenotype and an increase in tissue stiffness, indicating successful tendon-like differentiation without additional molecular or mechanical cues.

Overall, the manuscript is well-written, with a clear and informative presentation of the title, introduction, results, and discussion. The data and findings support the main conclusions; however, some aspects of the interpretation should be refined. Nevertheless, the study was a straightforward and engaging read, and the figures were well-organised and effectively presented.

I have provided several comments below aimed at improving the clarity, readability, and strength of the authors' conclusions.

Introduction

1. The first paragraph of the introduction is somewhat repetitive in relation to the following sections. I suggest the authors consider removing or revising this section to improve conciseness.
2. The authors state: "There is no identified master gene driving the tendon program. Moreover, developmental tendon fibroblast populations are not well characterized, and the molecular markers of tendon cell populations are not well identified (Hirsinger et al. 2024, <https://doi.org/10.1016/j.isci.2024.110305>)." However, Duprez and colleagues have characterised the tenogenic gene and cellular programs at single-cell RNA-seq resolution in the developing chicken limb. This statement seems counterintuitive, as there is substantial knowledge regarding developmental tendon fibroblast/progenitor populations. The authors also omit the relevant work of Arostegui et al. (Michael Underhill Lab, UBC, Canada; DOI: 10.1016/j.celrep.2023.112325). I strongly encourage the authors to integrate these findings into their discussion of tendon development.
3. The concept of a "master" gene regulating tendon differentiation is unclear. Gene expression is governed by a complex interplay of metabolic activity, transcription factors, chromatin modifications, and nucleosomal dynamics. The authors should clarify their definition of a "master" gene in this context. For instance, could *Hic1* qualify as such?
4. The manuscript refers to "mesenchymal stem cells," but it appears the authors are actually referring to mesenchymal stromal cells with progenitor-like properties. Clarification is needed.

Results and Figures

1. Page 7, line 182: The mention of apoptotic features lacks specificity. Please clarify what apoptotic markers were observed. Additionally, CASP3 alone may not be sufficient to rule out apoptosis. Have the authors considered a TUNEL assay? Apoptosis was assessed relatively late (days 7 and 14), meaning early apoptotic events may have been missed. Evaluating apoptosis at earlier time points (e.g., D0 or D2) would strengthen this analysis.
2. Page 7, line 197: The phrase "analyse" should be corrected to "analysis." Perhaps rewording as "We analysed commonly recognised..." would improve clarity.
3. Figure 3 and related datasets: A direct correlation between gene expression changes and increased stiffness in the 3D constructs is missing. This analysis would reinforce the authors' conclusions.
4. A key limitation of this study is the reliance on gene expression analysis (RT-qPCR and ISH) to evaluate tendon-like differentiation. For example, the authors state (page 9, lines 274-273): "We conclude that the 3D-hASC constructs acquire solid-like properties, consistent with the expression of matrix transcripts during the time course of the cultures." However, protein-level validation is lacking. Immunofluorescence staining or second harmonic generation microscopy could provide insight into collagen fibre organisation over time.
5. The authors focus primarily on the upregulation of tendon-associated genes. However, what about the loss of progenitor markers such as *Hic1* or *PDGFR α* ? This could provide additional evidence for a shift towards a tendon phenotype.

6. Have the authors assessed the proliferation potential of embedded cells within the 3D constructs? Direct assays such as EdU incorporation or thymidine analogues would provide valuable insights into cell proliferation.

Discussion

1. Page 11, first paragraph: As mentioned earlier, the manuscript does not establish a direct correlation between ECM/tendon-associated gene expression and the mechanical properties of the constructs. Additional analyses could address this gap.
2. Page 11, lines 325-326: "The molecular progression in 3D-hASCs is reminiscent of tendon gene expression during development." This conclusion is not fully supported by the presented data. The authors should temper their claims unless they conduct an unbiased transcriptomic analysis (e.g., RNA-seq) to directly compare their dataset with developmental tendon gene programs.
3. Page 11, lines 332-333: The phrase "Two newly identified genes" is vague. Does this refer to two genes identified within the study or within tendon-like gene programs in general? Please clarify.
4. The claim that cells segregate into two distinct populations lacks sufficient supporting evidence. The authors should either provide stronger data or refine this conclusion in the discussion.

Reviewer 2

Comments for the author

In their manuscript, Horde et al. characterized the differentiation of tendon cells by culturing the human adipose stromal cells (hASCs) embedded in collagen hydrogels using the Flexcell technology. In this 3D-hASCs system, the authors were able to track the expression of tendon-specific genes during a course of 3-week culture and detected an increase of those tendon marker genes, including transcriptional regulators and extracellular matrix genes, suggesting tension-specific differentiation. Further, by measuring the Young's modulus using nanoindentation, the authors discovered an increased stiffness of the 3D-hASCs culture during tension differentiation, consistent with the increased expression of extracellular matrix genes. Finally, by examine the organization of the cells, they found two cell populations that are formed at the center and the periphery, respectively, in the 3D-hASCs, reminiscent of the native tendon. Therefore, the authors demonstrate a powerful in vitro system to study the tendon development. The experiments were well designed, the results were neatly displayed, and the conclusions were convincing. The manuscript could be further improved if the authors could address the following points.

Major points:

1. As shown by the authors, the 3D-hASCs formed two cell populations at the center and at the periphery, respectively, and they show distinct gene expression. Do they have distinct material properties? Also, since this in vitro system allows the authors to directly monitor the process of the tension-specific differentiation, did the authors try to understand how the cell populations are formed? This could be important to reveal some mechanistic insights into tension formation that cannot be achieved through in vivo studies.
2. It is interesting that the differentiation of 3D-hASCs is tendon-specific, but not adipogenic or osteogenic, different from that of 2D cultures. Is there any explanation on why the 3D culture induced this specificity? The authors should discuss this.

Reviewer's Responses to Questions

Experimental quality

Does each figure have the proper controls?

If 'No', please indicate reasons in Comments for Author box below.

Reviewer #1:

- Yes

Reviewer #2:

- Yes
-

Were the data analyzed using appropriate statistical tests?

If 'No', please indicate reasons in Comments for Author box below.

Reviewer #1:

- Yes

Reviewer #2:

- Yes
-

Reproducibility

Were experiments performed using adequate number of biological replicates?

If 'No', please indicate reasons in Comments for Author box below.

Reviewer #1:

- Yes

Reviewer #2:

- Yes
-

Does the methods section provide sufficient detail to permit reproducibility?

If 'No', please indicate reasons in Comments for Author box below.

Reviewer #1:

- Yes

Reviewer #2:

- Yes
-

Completeness

Are the manuscript's conclusions supported by the data?

If 'No', please indicate reasons in Comments for Author box below.

Reviewer #1:

- Yes

Reviewer #2:

- Yes
-

Scholarship

Do the authors cite and discuss the merits of data that would argue for and against their conclusion?

If 'No', please indicate reasons in Comments for Author box below.

Reviewer #1:

- Yes

Reviewer #2:

- Yes
-

Does the manuscript title & abstract accurately reflect the contents of the manuscript, without hyperbole?

If 'No', please indicate reasons in Comments for Author box below.

Reviewer #1:

- Yes

Reviewer #2:

- Yes

First revision

Author response to reviewers' comments

Response to comments from the Reviewers:

Reviewer 1:

The manuscript by Horde et al., titled "Human adipose stromal cells differentiate towards a tendon phenotype with adapted visco-elastic properties in a 3D-culture system", investigates tendon cell differentiation in human adipose stromal cells (hASCs) using a minimal 3D system. By embedding hASCs in a type-I collagen gel with a static uniaxial constraint, the authors evaluated tendon-associated gene expression and mechanical properties over three weeks. The results demonstrate a molecular progression towards a tendon phenotype and an increase in tissue stiffness, indicating successful tendon-like differentiation without additional molecular or mechanical cues.

Overall, the manuscript is well-written, with a clear and informative presentation of the title, introduction, results, and discussion. The data and findings support the main conclusions; however, some aspects of the interpretation should be refined. Nevertheless, the study was a straightforward and engaging read, and the figures were well-organised and effectively presented.

I have provided several comments below aimed at improving the clarity, readability, and strength of the authors' conclusions.

Introduction

1. The first paragraph of the introduction is somewhat repetitive in relation to the following sections. I suggest the authors consider removing or revising this section to improve conciseness.

We agree with this comment and have removed the first paragraph of the introduction.

2. The authors state: "There is no identified master gene driving the tendon program. Moreover, developmental tendon fibroblast populations are not well characterized, and the molecular markers of tendon cell populations are not well identified (Hirsinger et al. 2024; <https://doi.org/10.1016/j.isci.2024.110305>)." However, Duprez and colleagues have characterised the tenogenic gene and cellular programs at single-cell RNA-seq resolution in the developing chicken limb. This statement seems counterintuitive, as there is substantial knowledge regarding developmental tendon fibroblast/progenitor populations.

We have modified our statement, we now state that "tendon fibroblast populations have been recently characterized" and cite relevant references about tendon fibroblast populations. The new sentence is: "*However, tendon fibroblast populations have been recently characterized with single-cell RNAsequencing technology in developing chicken and mouse limbs (Hirsinger et al., 2024; Arostegui et al., 2022; Coren et al., 2024).*" We have changed the text accordingly lines 66-68.

The authors also omit the relevant work of Arostegui et al. (Michael Underhill Lab, UBC, Canada; DOI:10.1016/j.celrep.2023.112325). I strongly encourage the authors to integrate these findings into their discussion of tendon development.

We have now integrated the relevant work of Arostegui et al. 2022 and 2023 in the introduction lines 66-68 and in the discussion lines 358 and 360.

3. The concept of a "master" gene regulating tendon differentiation is unclear. Gene expression is governed by a complex interplay of metabolic activity, transcription factors, chromatin modifications, and nucleosomal dynamics. The authors should clarify their definition of a "master" gene in this context. For instance, could *Hic1* qualify as such?

The definition of a master gene is to be necessary and sufficient to trigger a specific differentiation program from any cell types. *MYOD* is a master gene for the muscle lineage, since *MYOD* is able to trigger the muscle differentiation program in non-muscle cell in vitro and in vivo. Since *HIC1* is expressed in many connective tissue fibroblasts during development and in the adult, and is recognized to be a marker of the mesenchymal progenitor state during development, *HIC1* does not fulfil the criteria to be a master gene.

We have re-written this part in the introduction, lines 61-64. The new sentence is: "*There is no identified master gene for the tendon lineage, i.e. there is no identified gene whose expression alone can trigger the tendon differentiation programme; this stands in contrast to the controlling role of MYOD in the muscle programme (Tapscott, 2005).*" IBPS Dev2A SU-CNRS UMR 8263 Inserm 1345 7 Quai Saint-Bernard - Bâtiment C - Case 24 - 75252 Paris cedex 05 - France 3

4. The manuscript refers to "mesenchymal stem cells," but it appears the authors are actually referring to mesenchymal stromal cells with progenitor-like properties. Clarification is needed.

We agree with this comment and have changed the text accordingly, see lines 111-112. The new sentence is "*Mesenchymal Stromal Cells (MSCs) display progenitor-like properties and are capable of differentiating into different cell lineages, including osteoblasts, adipocytes or chondrocytes.*"

Results and Figures

1. Page 7, line 182: The mention of apoptotic features lacks specificity. Please clarify what apoptotic markers were observed. Additionally, CASP3 alone may not be sufficient to rule out apoptosis. Have the authors considered a TUNEL assay? Apoptosis was assessed relatively late (days 7 and 14), meaning early apoptotic events may have been missed. Evaluating apoptosis at earlier time points (e.g., D0 or D2) would strengthen this analysis.

We have now performed new cleaved-caspase 3 immunostaining at Day 0, Day 2, Day 4, Day 7, Day 14 and Day 21, which showed very few cleaved-caspase-positive cells. We conclude that very little Caspase3-apoptosis was observed in these experiments. This result is consistent with the relatively little change in cell number in 3D hASCs between Day 0 and Day 21 (Fig. 1D). This result is now presented in new Fig. S2 and in the text line 191.

2. Page 7, line 197: The phrase "analyse" should be corrected to "analysis." Perhaps rewording as "We analysed commonly recognised..." would improve clarity.

We have modified the text accordingly. The new sentence is: "*We first analysed recognised tendon markers at different time points during the culture*", lines 209-210.

3. Figure 3 and related datasets: A direct correlation between gene expression changes and increased stiffness in the 3D constructs is missing. This analysis would reinforce the authors' conclusions.

We agree that our data show a temporal correlation between gene expression changes and increased stiffness in the 3D-constructs. Establishing a direct correlation between the expression of

tendon-associated genes and the mechanical properties of 3D hASCs will require further functional experiments and will be the subject of another study.

4. A key limitation of this study is the reliance on gene expression analysis (RT-qPCR and ISH) to evaluate tendon-like differentiation. For example, the authors state (page 9, lines 274-273): "We conclude that the 3D-hASC constructs acquire solid-like properties, consistent with the expression of matrix transcripts during the time course of the cultures." However, protein-level validation is lacking. Immunofluorescence staining or second harmonic generation microscopy could provide insight into collagen fibre organisation over time.

To complement the gene expression analysis by RT-qPCR and ISH experiments, we have performed immunostaining with collagen type I antibody in 3D-hASCs at Day 0, Day 2, Day 4, Day 7, Day 14 and Day 21. This shows the orientation of the collagen fibres along the axis of the constructs. The results are presented in new Fig. S4 and in the results lines 202-203.

5. The authors focus primarily on the upregulation of tendon-associated genes. However, what about the loss of progenitor markers such as *Hic1* or *PDGFR α* ? This could provide additional evidence for a shift towards a tendon phenotype.

These progenitor markers, *HIC1* and *PDGFR α* , are also expressed in the tendon lineage, making it difficult to draw conclusion from their expression levels along the tendon differentiation process.

Nevertheless, we analysed their expression in 3D-hASCs over time using RT-qPCR. In addition, we analysed the expression of *PRRX1*, another marker of mesenchymal stem cell progenitors, which is also expressed in developing tendons (Hirsinger et al, 2024). We did not observe an obvious decrease in the expression of these three markers in 3D-hASCs over time (we even observed an increase in expression at different time points). We believe that the peak of *HIC1* expression at Day 4 suggests that the hASCs are undergoing a teno-chondrogenic progenitor step (Arostegui, Scott and Underhill, 2023). However, the hASCs in 3D cultures favour a tendon fate, as cartilage markers are not expressed in 3D-hASC constructs over time.

These results are now included in the results (lines 226-233) and discussed in the discussion section (lines 356-360).

IBPS Dev2A SU-CNRS UMR 8263 Inserm 1345 7 Quai Saint-Bernard - Bâtiment C - Case 24 - 75252 Paris cedex 05 - France 4

6. Have the authors assessed the proliferation potential of embedded cells within the 3D constructs?

Direct assays such as EdU incorporation or thymidine analogues would provide valuable insights into cell proliferation.

To assess proliferation, we have now performed phospho-histone H3 immunostaining in 3D-hASCs at Day 0, Day 2, Day 4, Day 7, Day 14 and Day 21, which showed very few PH3-positive cells. This result is consistent with the relatively constant number of cells in 3D-hASCs between Day 0 and Day 21 (Fig. 1D). These results are now presented in a new Fig. S3 and in the Results, lines 191-192.

Discussion

1. Page 11, first paragraph: As mentioned earlier, the manuscript does not establish a direct correlation between ECM/tendon-associated gene expression and the mechanical properties of the constructs. Additional analyses could address this gap.

As mentioned above, we have established a temporal correlation between ECM/tendon gene expression. We mention in the discussion that the stiffness and solid-like properties of 3D-hASCs parallel the expression of tendon matrix genes. We have not established a direct correlation between ECM/tendon-related genes and the mechanical properties of 3D-hASCs based on the data presented in this article. We discuss a direct correlation as a perspective in the discussion section lines 387-390.

2. Page 11, lines 325-326: "The molecular progression in 3D-hASCs is reminiscent of tendon gene expression during development." This conclusion is not fully supported by the presented data. The authors should temper their claims unless they conduct an unbiased transcriptomic analysis (e.g., RNA-seq) to directly compare their dataset with developmental tendon gene programs.

We agree with this comment and we have rewritten the sentence. The new sentence is: "*The expression of the developmental tendon genes, SCX, MKX and TNMD in 3D-hASCs is reminiscent of their molecular progression during development*", lines 334-335.

3. Page 11, lines 332-333: The phrase "Two newly identified genes" is vague. Does this refer to two genes identified within the study or within tendon-like gene programs in general? Please clarify.

We have clarified this point lines 342-344. These "*newly identified genes*", *TM4SF1 and ANXA1, were identified in the Top100 genes (ordered with the highest enrichment score) of SCX-positive cells during mouse limb development*".

4. The claim that cells segregate into two distinct populations lacks sufficient supporting evidence. The authors should either provide stronger data or refine this conclusion in the discussion.

We observed that hASCs in 3D cultures self-organise into two populations, one in the centre and one at the surface. This behaviour has been already observed in other 3D cell cultures (Yang et al., 2013; Garvin et al., 2003). In addition, we found that peripheral cells lost *COL1A1* expression at Day 21, whereas it was maintained in central cells, indicating a difference in gene expression between the two cell populations.

We have tried to better explain the cell organisation in 3D cultures over time and have modified the text, lines 297-302 and in the discussion lines 398-413 accordingly.

IBPS Dev2A SU-CNRS UMR 8263 Inserm 1345

7 Quai Saint-Bernard - Bâtiment C - Case 24 - 75252 Paris cedex 05 - France 5

Reviewer 2:

In their manuscript, Horde et al. characterized the differentiation of tendon cells by culturing the human adipose stromal cells (hASCs) embedded in collagen hydrogels using the Flexcell technology. In this 3D-hASCs system, the authors were able to track the expression of tendon-specific genes during a course of 3-week culture and detected an increase of those tendon marker genes, including transcriptional regulators and extracellular matrix genes, suggesting tension-specific differentiation.

Further, by measuring the Young's modulus using nanoindentation, the authors discovered an increased stiffness of the 3D-hASCs culture during tension differentiation, consistent with the increased expression of extracellular matrix genes. Finally, by examine the organization of the cells, they found two cell populations that are formed at the center and the periphery, respectively, in the 3DhASCs, reminiscent of the native tendon. Therefore, the authors demonstrate a powerful in vitro system to study the tendon development. The experiments were well designed, the results were neatly displayed, and the conclusions were convincing. The manuscript could be further improved if the authors could address the following points.

Major points:

1. As shown by the authors, the 3D-hASCs formed two cell populations at the center and at the periphery, respectively, and they show distinct gene expression. Do they have distinct material properties? Also, since this in vitro system allows the authors to directly monitor the process of the tension-specific differentiation, did the authors try to understand how the cell populations are formed?

This could be important to reveal some mechanistic insights into tension formation that cannot be achieved through in vivo studies.

Given the composite and anisotropic structure of our samples, it is difficult to predict the stress field (i.e. how mechanical stress propagates within the volume of the sample) within the construct during nanoindentation. Our experiments (using probe radii of 25 to 50 μm) are likely to probe a mixture of the two cell populations.

The tension-specific influence on the formation of the two cell populations was not addressed in this manuscript. This will be the subject of another project.

2. It is interesting that the differentiation of 3D-hASCs is tendon-specific, but not adipogenic or osteogenic, different from that of 2D cultures. Is there any explanation on why the 3D culture induced this specificity? The authors should discuss this.

The analysis of the lineage potential in 2D-cultures was to show that the hASCs were able to differentiate towards the adipogenic and osteogenic lineages. The fact that hASCs do not go towards these lineages when embedded in collagen hydrogel maintained between two anchor points shows that these 3D-culture conditions favor tendon differentiation. This is better explained in the results lines 233-237.

Second decision letter

MS ID#: bio.061911R1

MS TITLE: Human adipose stromal cells differentiate towards a tendon phenotype with adapted visco-elastic properties in a 3D-culture system

AUTHORS: Maxime Horde; Jonathan Fouchard; Luna Gomez Palacios; Xavier Laffray; Cedrine Blavet; Veronique Bereziat; Claire Lagathu; Ludovic Gaut; Delphine Duprez; Emmanuelle Havis

Thank you for addressing the reviewers' comments to the best of your ability. I am happy to tell you that your manuscript has been accepted for publication in Biology Open, pending our standard publication integrity checks. It was accepted on 14 Apr 2025.